# Short-Term CO_2_ Treatment of Harvested Grapes (*Vitis vinifera* L., cv. Trebbiano) before Partial Dehydration Affects Berry Secondary Metabolism and the Aromatic Profile of the Resulting Wine

**DOI:** 10.3390/plants11151973

**Published:** 2022-07-29

**Authors:** Marco Santin, Stefano Brizzolara, Antonella Castagna, Annamaria Ranieri, Pietro Tonutti

**Affiliations:** 1Department of Agricultural, Food and Agro-Environmental Sciences (DAFE), University of Pisa, Via del Borghetto 80, 56100 Pisa, Italy; marco.santin@unipi.it (M.S.); antonella.castagna@unipi.it (A.C.); 2Crop Science Research Center, Scuola Superiore Sant’Anna, Piazza Martiri della Libertà 33, 56127 Pisa, Italy; stefano.brizzolara@santannapisa.it (S.B.); pietro.tonutti@santannapisa.it (P.T.)

**Keywords:** carbon dioxide, postharvest dehydration, carotenoids, free and glycosylated volatiles, phenols, secondary metabolism

## Abstract

High CO_2_ concentrations applied to harvested horticultural products can modify primary and secondary metabolism. This work reports the metabolic responses to short-term CO_2_ treatments of white-skinned grapes (cv Trebbiano) undergoing postharvest partial dehydration. The influence of CO_2_ treatments on the aroma profile of the derived sweet wine was also assessed. Harvested grapes were treated with gaseous CO_2_ (30%) or air (control) for 24 h and then dehydrated (about 45% of weight loss) before vinification. Lipophilic and phenolic compounds of grape skin and the wine aroma profile were analyzed. In CO_2_-treated berries, the lipophilic and phenolic compounds decreased at a reduced and faster rate, respectively, during dehydration. Aroma profile of wine from CO_2_-treated grapes showed a slight but significantly higher content of glycosylated C13 and terpene compounds, and a decrease/absence of free acids, vanillin derivates and other phenol volatiles. The higher content of volatile alcohols in wine from treated berries suggests that the alcoholic fermentation was triggered. CO_2_ application before the withering process of Trebbiano grapes affects the aroma profile of the resulting wine by altering the free:glycosylated volatiles ratio. This study provides information on the possible use of CO_2_ as metabolic elicitor to modulate the aroma profile of the resulting wines obtained after grape dehydration.

## 1. Introduction

Harvested fruits reacts to the imposed environmental conditions (temperature, atmosphere composition) by altering the metabolism and the metabolic profile. Together with the decrease in oxygen concentration, the increase in carbon dioxide levels characterizes the controlled atmosphere (CA) protocols applied for prolonging the commercial life of different fruit species such as apples, pears, and kiwifruit. It is well known that elevated CO_2_ concentrations affect primary and secondary metabolism of harvested products, and the responses may be beneficial or detrimental depending upon the nature of the product, the concentration of CO_2_ outside and inside the tissue, the duration of exposure and the concentration of oxygen [1,2,3]. In addition to the more or less prolonged storage protocols under high CO_2_ concentration (and different low levels of oxygen), much evidence shows that temporary/short-term (from few hours to some days) treatments with high CO_2_ are effective in eliciting specific metabolic responses of fruit tissues in the subsequent shelf/storage life. For example, short-term high-CO_2_ treatments were effective in reducing astringency in persimmons [4], maintaining physicochemical, microbial, and sensory quality of strawberries [5], and limiting decay and insect infestation, as observed in table grapes [6]. Considering specifically grapes, short-term treatments (up to 72 h) with high CO_2_ (20–30%) immediately after harvest reduced the total fungal decay and induced a lower rate of water loss and browning index in comparison with control samples during the subsequent storage [7]. These short-term treatments also affected grape berry secondary metabolic processes such as those related to polyphenols, resulting in altered phenylalanine ammonia lyase (PAL), stilbene synthase (STS), and chalcone synthase (CHS) activities [8], and inducing transient increases of flavan-3-ols accumulation in the skin [9]. In addition, effects on oxidative stress, plasma membrane and cell wall have been reported as a result of short-term high CO_2_ treatments on grapes [10]. The treatments with high CO_2_ induce marked changes in the expression of several structural and regulatory genes directly or indirectly involved in different quality-related metabolic processes, including those affecting the aroma profiles of the grape berries [9]. Detailed studies on the effect on aroma profile induced by high CO_2_ concentration treatment applied to persimmon to induce deastringency have been reported [11,12]. According to the published literature [5,8,9,10,11], some of the effects of the short-term treatments with high CO_2_ concentrations on harvested grapes can be detected soon after the end of the treatment, while others are present also after several days or weeks following the application of the enriched carbon dioxide atmosphere.

The effectiveness of short-term high CO_2_ treatments in reducing the incidence of decay (due to *Botrytis*, in particular) and in modulating the grape berry metabolism during storage may be of interest for the wine industry, in particular for the production of specific wine types (e.g., Vin Santo, Amarone) for which harvested grapes undergo a more or less prolonged dehydration process before vinification [13]. Besides the concentration effect, the grape dehydration process (for some wine productions up to 50–60% of weigh loss) induces changes of the overall metabolism, including those related to the aroma profile that makes these wine types unique [14,15]. In fact, several changes in aroma composition, which also involve the accumulation of C6 compounds, ethanol, ethyl acetate, terpenes and norisoprenoids, take place during the more or less prolonged dehydration processes [16,17,18,19]. Taking into consideration the recognized effect as elicitor of carbon dioxide, in the present paper, we studied the impact of a short-term high CO_2_ treatment (30%, for 24 h) on specific compositional parameters of white-skinned cv Trebbiano grapes undergoing prolonged dehydration (up to about 45% of weight loss), and the effect of such treatment on the aromatic profile of the resulting wine.

## 2. Results

Weight loss of treated and control Trebbiano grapes was monitored throughout the experimental period (up to 73 days in the dehydration chamber). Although no statistically significant differences were detected, CO_2_-treated grapes showed a lower dehydration value than control after the first 15 days of the process, followed by a faster dehydration trend during the remaining period till the end of the trial, when control and CO_2_-treated samples lost 45.78 and 47.08% of fresh weight, respectively (Table 1).

### 2.1. Lipophilic and Phenolic Compounds in Berry Skins

At harvest time, ß-carotene and chlorophylls *a* and *b* were the most abundant lipophilic compounds in skin tissue of Trebbiano grape berries (Appendix A). When berries lost about 30% of weight (T1, Figure 1A), ß-carotene, violaxanthin, chlorophyll *a* and chlorophyll *b* were significantly lower in the skin of control compared to CO_2_-treated samples. These samples displayed concentrations of these compounds comparable to those detected at harvest (T0). Lutein, neoxanthin, zeaxanthin and antheraxanthin showed a similar content between treated and control grapes. With the exception of lutein, the other three carotenoids decreased in both control and CO_2_-treated samples if compared to T0. At the end of storage (T2, Figure 1A), the treatment did not generally affect the lipophilic compounds, except for chlorophyll *b* that showed a greater content in CO_2_-treated skin berries in respect to the control. Again, except for lutein, all quantified lipophilic compounds decreased at T2, in both control and CO_2_-treated samples, compared to T0.

Total flavonoids level was affected by the CO_2_ treatment, with a significant lower amount recorded in the CO_2_-treated samples at both T1 and T2, compared to control (Figure 1B). Differently, total phenols, flavonols, tartaric acid esters, flavan-3-ols and condensed tannins (proanthocyanidins) did not show any significant difference between control and CO_2_-treated samples, regardless of the sampling time (Figure 1B and Appendix A). However, considering the concentration values of the whole set of phenol compounds quantified, it is interesting to note that, compared to T0, at T1, a general increasing trend is present, followed at T2 by a decrease.

### 2.2. Free and Glycosylated Aroma Compounds in Trebbiano Wines

The wines produced from CO_2_-treated and untreated berries did not differ in terms of technological parameters (alcohol content, pH, total acidity; data not shown)

The aroma profile of wine was analyzed measuring both free and glycosylated compounds. A total of 56 free aroma compounds and 41 glycosylated molecules have been identified. Considering the overall profile and regardless of the treatment, free volatile organic compounds (VOCs) belonging to acids, alcohols, esters, norisoprenoids (C13) and volatile phenols chemical classes were the most represented (Appendix A). On the other hand, alcohols, norisoprenoids (C13), volatile phenols, and terpenes were the most represented classes of glycosylated aroma compounds detected in Trebbiano wine (Appendix A).

Among free volatiles, 15 compounds, namely palmitic acid, ethyl-9-decanoate, diethyl malate, ethyl-c18, ethyl oleate, ethyl linoleate, pentalactone, 3-oxo-α-ionol, vanillin, methyl vanillate, ethyl vanillate, acetovanillone, vanillylacetone, guaiacyl propanol, guaiacyl ethanol, have been found only in wines produced from control samples. On the other hand, 3 glycosylated aroma compounds have been detected only in wines from treated grapes, namely cis- and trans-LOF (linalool oxide furanoid), and the signal assigned to 3,8-diol + nerol hydrate. These results suggest that the treatment with CO_2_ could modulate the ratio between free and glycosylated molecules, by inducing such a strong decrease in several free compounds to make them nondetectable any more after the treatment, and a parallel ex novo production of some glycosylated molecules which are absent in control wines. Free volatile phenols were the most affected chemical class. Seven of the eight identified phenols were nondetectable in wines from CO_2_-treated grapes, 4-vinyl guaiacol being the only molecule present in both samples (Appendix A). In addition, a marked reduction in decanoic acid (capric acid), together with the disappearance of hexadecanoic acid (palmitic acid), characterized the wine obtained from CO_2_-treated grapes. Interestingly, and regardless of the treatments, terpenes have been detected only in the glycosylated form, while dioxanes/dioxolanes and acids families were present only among the free identified aroma compounds. N-ethylacetamide was the only dioxanes derivative detected, and its concentration did not vary between CO_2_-treated and control wines. Among glycosylated esters, only ethyl caprate and isopropyl myristate were detected in Trebbiano wines, and they showed the same content between control and CO_2_-treated wines (Appendix A).

Free and glycosylated volatiles have been employed to perform a Partial Least Squares Discriminant Analysis (PLSDA; Figure 2), for which only the molecules that contributed the most to control and treated samples clustering have been included considering the VOCs that were commonly identified in both wine samples. The PLSDA model explains a remarkably high percentage of the variability present in the dataset (about 98% considering both first and second factor) and displays a clear clustering of the different samples, with control wines located on the left quadrants of the plot and wines from CO_2_-treated grapes on the opposite side. Control samples appeared to be less homogeneous than treated samples, resulting more scattered on the plot (Figure 2).

The PLSDA analysis highlights that several free aromas, or signals related to them (e.g., butyrolactone + acid C4), are associated with control wines, namely decanoic acid, ethyl decanoate, ethyl C16, ethyl hexadecanoate, methionol and 4-carbethoxy-butyrolactone precursor. On the other hand, Figure 2 also shows that other free molecules, namely 2,3-butanediol, 1,2-propanediol, diethyl succinate and 1,3-propanediol monoacetate, and glycosylated compounds, such as linalool, alpha-terpineol, citronellol hydrate, (Z)-8-hydroxylinalool + geraniol hydrate, and 3-hydroxy-7,8-dihydro-β-ionol, were found at higher levels in wines from CO_2_-treated grapes.

## 3. Discussion

In the wine industry, carbon dioxide is used for two main reasons: applying carbonic maceration (CM) for Beaujolais wine style, and rapidly cooling grape or must [20]. CM is aimed to produce light- to medium-bodied red wines to make them fruitier and to soften their tannins. According to this protocol, intact bunches of grapes are incubated for several days in sealed vessels with carbon dioxide, and this oxygen-free environment induces the start of intracellular (pre)fermentation not driven by yeasts. In the present work, a different approach is used, in which CO_2_ is applied to modulate berry metabolism and composition during the dehydration process. The off-vine dehydration is receiving increasing attention and interest not only to produce dessert/sweet wines (such as the “passiti”), but also dry and full-bodied red wines in the style of “Amarone” [13]. The dehydration process carried out in specialized facilities (tunnels, chambers) allows not only the management of the environmental parameters (temperature, relative humidity—RH, ventilation), but also the application of specific treatments that may have an effect on controlling the pathogen (fungi) attack, may affect the water loss rate and elicit specific metabolic responses of the berries. In these production protocols, the rate and the intensity of dehydration play a key role in modulating the physiological responses of the berries to the applied stress, with marked consequences in terms of general metabolism and composition, and, hence, in specific traits of the resulting wines. Previous works reported an effect of short-term treatments with high CO_2_ concentrations in reducing water loss in harvested table grapes [7,10,21,22,23]. Our data confirm in part these observations, pointing out that, if an effect on weight loss is induced by the CO_2_ treatment, this is present only within the initial 15 days of dehydration. Blanch et al. [24] observed that short-high CO_2_ treatments were beneficial for preventing water loss in strawberry, possibly due to an increase in cellular water retention associated with an accumulation of osmolytes. If this occurs also in grape berries, it remains to be elucidated.

In general, high CO_2_ concentrations applied on harvested mature fruits may have the effect of delaying ripening. This effect can be observed also in the present trial considering the changes in lipophilic compounds, in particular the concentration of chlorophylls *a* and *b* and ß-carotene that, at T1, was higher than in control samples and similar to that detected at harvest (T0). Chlorophyll *a* and *b* as well carotenoids markedly decrease in grape berries during the last stages of development, reaching the lowest values in correspondence of ripening, as observed in sixteen white wine and table grape cultivars by Rocchi et al. [25]. This decreasing trend has also been detected in our control samples from T0 to T1 and from T1 to T2. The delay of this process by CO_2_ observed at T1 might be due to the effects of the gas on the enzymes responsible for the degradation of these pigments. Huang et al. [26] demonstrated that high-CO_2_ treatment in Chinese cabbage decreased the activity of chlorophyllase and lowered the transcript abundances of specific genes (BrChlase, BrPAO, and BrRCCP) responsible for chlorophyll degradation. Except for chlorophyll *b*, the observed effect of CO_2_ on lipophilic compounds at T1 was not detected in the Trebbiano grape skins at T2.

To the best of our knowledge, no information is available in the literature about the relationship between the postharvest application of CO_2_ on grape berries and its effect on wine volatile profile. One of the major alterations we found in wines obtained from CO_2_-treated berries is a strong reduction in free volatile phenols, such as vanillin, methyl vanillate, ethyl vanillate, acetovanillone, vanillylacetone, guaiacyl propanol, and guaiacyl ethanol. A marked reduction in specific volatile phenols (namely 4-ethylguaiacol, 4-ethylphenol, guaiacol) was reported by Gonçalves et al. [27] in grape berries grown under high level of CO_2_. Moreover, CM induces in wines a significant decrease in acetovanillone and 4-vynil guaiacol, together with a significant increase in ethyl vanillate [28,29]. Although these papers refer to different protocols and methods, these results may indicate that CO_2_ is effective in modulating volatile phenol metabolism in grape berries.

Other important modifications observed in VOCs profile of wines obtained from CO_2_-treated grape are linked to the reduction and/or the nondetection of specific free acids (decanoic and hexadecenoic, associated with negative descriptors, such as sweat, rancid, fat), ethyl esters (decanoate, 9-decanoate, γ-hexadecanoate, oleate, linoleate, C16 and C18), together with an increase in diethyl succinate and 1,3 propanediol monoacetate. Marked differences in ethyl derivatives have also been detected in wine from grapes grown under high CO_2_ concentration, with significantly higher levels of ethyl octanoate, 2-methylbutyl acetate and hexanoate, and reduced amounts of ethyl acetate [27]. When investigating grape composition after high CO_2_ storage, Dourtoglou et al. [30] reported increased levels of ethyl 2-butenoate, hexanoate, octanoate, salicilate, dodecanoate and cinammate. Considering the class of esters, highly impacting the overall aroma of a sweet wine, no specific trend is observed, with some free compounds increased and others decreased in wine from CO_2_-treated grapes. Published papers on CM effect on wine aroma also show contrasting results regarding esters, also depending on the investigated grape variety and the specific protocol applied [28,29].

A significant decrease in butyrolactone derivatives (γ-butyrolactone + acid c4 and 4-carbethoxy-γ-butyrolactone precursor) and methionol has been observed in wine from CO_2_-treated grapes. Considering lactones, in the literature, different results are reported, with a significant increase in γ-decalactone and γ-undecalactone in grapes stored under high CO_2_, and a significant decrease in butyrolactone in wines after carbonic maceration protocol [28,30]. Considering specifically methionol, Gonçalves et al. [27] reported similar results in wine obtained by grapes grown under high CO_2_.

Considering alcohols, a significant increase in specific molecules and their derivatives, namely 2,3-butanediol, 1,2-propanediol and 1,3 propanediol monoacetate, has been observed in Trebbiano wines obtained by CO_2_-treated grapes (Figure 2). Interestingly, 2,3 butanediol significantly increased also in wine obtained applying CM [31].

Several glycosylated norisoprenoids (3-hydroxy-7,8-dihydro-β-ionol, 3-oxo-7,8-dihydro-α-ionol), as well as other monoterpenoids (α-terpineol, linalool, cis- and trans- linalool oxide furanoid, citronellol hydrate, 3,8-diol + nerol hydrate and (Z)-8-hydroxylinalool + geraniol hydrate), significantly increased in wine from CO_2_-treated grapes. In general, terpenoids concentration has a very profound impact on wine aroma, given the very low odor threshold they often possess, and they can be interconverted during fermentation due to oxidation processes. Similarly, different terpenoids have been found to significantly increase in wine produced from grapes grown under high CO_2_ levels [27] and grapes stored under high CO_2_ [30].

## 4. Conclusions

In conclusion, our data indicate that harvested Trebbiano grapes react to short-term high CO_2_ treatment by altering some ripening/senescence-related processes during postharvest withering, with specific effects also induced on the volatile metabolism, differently impacting the VOC classes, and the free vs glycosylated aroma compound balance. Whether these changes are affecting positively, or negatively, wine taste and aroma still needs to be elucidated. Further investigations are needed to unravel the effect of pre-processing CO_2_ treatment on the organoleptic quality of Trebbiano wine and to better characterize the relationship between the changes induced in the berries and those detected in the final wine.

## 5. Materials and Methods

### 5.1. Experimental Plan

Grapes (*Vitis vinifera* L., cv ‘Trebbiano’, white-skinned berries) were manually harvested in correspondence of about 18°Bx (Brix) at Fattoria Fibbiano, Terricciola (Pisa, Tuscany, Italy). Selected bunches (for a total of about 100 kg) were placed inside a dehydration tunnel (farm scale) to lose water, and CO_2_ (30%) was fluxed at the rate of about 1.5 L h^−1^ for 24 h at 14 °C. At the end of the treatment, the tunnel was opened, and additional 100 kg of grapes (control) were loaded and both CO_2_-treated and control grapes were kept at 14 °C and 60% relative humidity (RH) for 73 days, when about 45% of fresh weight was lost. The weight loss dynamic was kept monitored throughout the experiment.

At T0 (harvest) and when berries lost about 30% (T1) and 45% (T2, end of the trial) of fresh weight, representative berry samples from different bunches (for a total of 100 berries) were collected and skins were frozen in liquid nitrogen. At the end of the experiment (73 days of dehydration), treated and control grapes were used to make wine by applying the standard protocols used in the winery for the production of “Trebbiano passito” wine, with no selected wine yeast strains added. After 6 months from vinification, wines obtained from control and CO_2_-treated grapes were analyzed in terms of technological parameters and aroma profiles (free and glycosylated compounds).

### 5.2. Berry Lipophilic Compounds Extraction and HPLC Detection and Quantification

Skin samples were pooled and ground using liquid nitrogen. A total of 0.7 g of powder for each biological replicate was homogenized in the dark in 100% HPLC-grade acetone with 1 mM sodium ascorbate, then filtered through 0.2 µm filters (Minisart SRP 25, PTFE-membrane) according to Castagna et al. [32]. The final volume was reported, and the extract (10 µL) was injected into an HPLC equipment. The analytical separation of lipophilic compounds was performed using Spectra SYSTEM P4000 HPLC equipped with a UV 6000 LP photodiode array detector (Thermo Fisher Scientific, Waltham, MA, USA). A Zorbax ODS column (Chrompack SA, 5µm particle size, 250 × 4.60 mm) was used. The mobile phase flow rate was set at 1.0 mL min^−1^. The gradient elution was conducted using methanol (25%) and acetonitrile (75%) (A), and methanol (68%) and ethyl acetate (32%) (B) as mobile phases: 0–10 min, 100% A, 0% B; 12–24 min, 0% A, 100% B; 26–32 min, 100% A, 0% B. The detection has been performed at 445 nm. The identification and quantification were carried out using UV spectra, peak retention times and peak areas from calibration curves established using external standards. The analysis has been performed on three biological samples for each treatment and the results have been expressed as µg/g of fresh weight.

### 5.3. Berry Polyphenol Extraction and Quantification

Skin samples were pooled and ground using liquid nitrogen. A total of 0.5 g of powder for each biological replicate was used, and extraction was carried out three times on a magnetic stirrer using a total volume of 90 mL of methanol:water (80:20, *v*:*v*). The liquid extracts were separated by centrifugation (14,000× *g*, 15 min) at 4 °C. Extracts, reduced to 16 mL using a rotary evaporator, were filtered with 0.45 µm filters Minisart, aliquoted (8 mL each aliquot) and stored at −80 °C.

Total phenols were determined using the Folin–Ciocalteu method, modified as described by Borbalan et al. [33]. A total of 1.85 mL distilled water, 0.125 mL Folin–Ciocalteu reagent and 0.5 mL 20% sodium carbonate solution were added to 25 µL liquid extract in a test tube (2.5 mL final volume). The solution was homogenized, and absorbance was recorded at 750 nm after 30 min. Concentration of total phenols was expressed as mg of gallic acid equivalents. Total flavonoids were determined as described by Kim et al. [34]. Moreover, 60 µL 5% NaNO_2_, 40 µL 10% AlCl_3_ and 400 µL 1 M NaOH were added to 100 µL liquid extract. The solution was diluted with 200 µL distilled water, mixed, and absorbance was recorded at 510 nm. Flavonoids concentration was expressed as mg of catechin equivalents. Tartaric acid esters and flavonols contents were determined according to Romani et al. [35]. A total of 25 µL of sample extract was diluted with 225 µL 10% ethanol and 250 µL 0.1% HCl in 95% ethanol, and 1 mL of 2% HCl was added. The solution was mixed and the absorbances were recorded at 320 nm for tartaric acid esters, and 360 nm for flavonols. Tartaric acid esters and flavonols concentrations were expressed as mg of caffeic acid and quercetin, respectively. Total flavan-3-ols were determined with p-(dimethylamino)-cinnamaldehyde (DMACA) reagent, as described by Nagel et al. [36]. A total of 10 µL of liquid extract was diluted with 90 µL methanol. Next, 250 µL 0.24 N HCl in methanol, 250 µL DMACA solution (0.2% in methanol) and 250 µL methanol were added. The absorbance was recorded at 640 nm and the concentration of flavan-3-ols was expressed as mg of catechin equivalents. Condensed tannins, or proanthocyanidins, were determined in accordance with the method described by Waterman et al. (1994). Butanol reagent was prepared by mixing 128 mg FeSO_4_ 7H_2_O with 5 mL concentrated HCl and brought to a final volume of 100 mL with n-butanol. A total of 50 µL of liquid extract was mixed with 700 µL butanol reagent and heated at 95 °C in a water bath for 45 min. Once the samples were cooled down, 250 µL n-butanol was added and the absorbance was measured at 550 nm. The total amount of condensed tannin was expressed as mg of cyanidin equivalents. The assays were performed using Ultrospec 2100 pro UV/Visible Spectrophotometer (Amersham BioSciences, Buckinghamshire, UK). The analysis has been performed on three biological replicates for each treatment.

Results of phenols, flavonoids, tartaric acid esters, flavonols, flavan-3-ols and proanthocyanidins are expressed per g of dry weight (Appendix A).

### 5.4. Aroma Compounds Extraction and Analysis of the Wines

#### 5.4.1. Total Glycosylated Aroma Compounds Extraction

To isolate glycosidically-bound aromatic compounds, wine samples (100 mL for each treatment) were stirred for 20 min with PVPP and water and then filtered under vacuum with a nitrocellulose Whatman filter. The filtered wine was eluted with Lichrolut RP-18 cartridges (43–63 μm, 500 mg 3 mL, Standard PP-Rohrchen and PP-tubes by Merk) at a flux rate of 3 mL min^−1^ (Autotrace SPE Workstation). The employed mobile phases were water, pentane:dichloromethane (2:1) and methanol. Before enzymatic hydrolysis, samples recovered in methanol were kept at −20 °C [37].

#### 5.4.2. Enzymatic Hydrolysis of the Total Glycosylated Aroma Compounds

Wine extracts were vaporized (Turbo VAP LV Concentration Workstation) with gaseous nitrogen for two hours and resuspended in 2 mL citrate-phosphate buffer (0.2 M, pH 5.0). Samples were added with 100 μL of AR200 enzymatic solution (70 mg mL^−1^, pH 5.0) and enzymatically hydrolyzed overnight at 40 °C. Hydrolyzed samples were extracted with pentane:dichloromethane (2:1) to recover the free aglycons, and the organic layer was filtered using glass wall and dried over Na2SO4. This procedure was repeated five times. An internal standard (nonanol, 3.22 mg mL^−1^) was added to the samples. Samples were concentrated at 35 °C (Turbo VAP LV Concentration Workstation) using Dufton columns and kept at −20 °C until GC-MS analysis [37].

#### 5.4.3. Total Free Aroma Compounds Extraction

Free aroma compounds extraction was carried out in triplicate according to Selli et al. [38] with the following modification: 100 mL of wine was added with 50 mL dichloromethane and 10 μL of nonanol (3.22 mg mL^−1^) as internal standard, and then stirred (30 min, 4 °C) and centrifuged (20 min, 9000× *g*, 4 °C). This procedure was repeated twice. The organic phases were collected into bottles and dried over sodium sulphate. Extracted samples were distilled at 45 °C (LABOMECA) and then kept at −20 °C prior to GC-MS analysis.

#### 5.4.4. GC-MS Analysis

GC-MS analysis of volatiles was performed using an Agilent 6890 Series System gas chromatograph, coupled to a HP Hewlett Packard 5973 Mass Selective Detector equipped with an AOC-5000 Auto Injector Shimadzu and a fused capillary column (DB-Wax, 60 m × 0.250 mm i.d., 0.25 μm film thickness). The injector temperature was set from 20 to 250 °C at 180 °C min^−1^ (2 min), then held at 250 °C for 80 min. The column temperature program was the following: 60 °C for 3 min; from 60 to 220 °C at 2 °C min^−1^; from 220 to 245 °C at 3 °C min^−1^ and then held 20 min at 245 °C. The flow of helium (carrier gas) was 1.5 mL min^−1^. The transfer line temperature was 250 °C, with source temperature of 250 °C; quadrupole temperature was set at 150 °C. Mass Spectra (MS) were scanned in the range m/e 29–350 amu employing intervals of 1 s. Identification of the compounds was performed by comparing linear retention index and electronic mass spectra with published data or reference standards. The levels of the volatile compounds were expressed as nonanol equivalents. Aroma analyses of the wine samples have been performed in three technical replicates for each treatment.

### 5.5. Statistical Analyses

Values reported in Figure 1 are the results of a fold change analysis performed applying the following formula: FC = log_2_ [replicate/mean (T0)], with carotenoid, chlorophyll, xanthophyll, and phenol content expressed as fold change of their level in correspondence of weight loss of about 30% (T1) and at the end of the storage after 73 days (T2) normalized on their level at harvest (T0). The effect of CO_2_ treatment has been evaluated applying, after testing its assumptions (Shapiro–Wilk test), one-way analysis of variance, ANOVA (*p ≤* 0.05), and *t*-test (*p ≤* 0.05) analysis (Table 1 and Figure 1, respectively). Moreover, Partial Least Squares Discriminant Analysis (PLSDA) model have been created to investigate treatment effect on free and glycosylated aromatic compounds. Variable in projection scores (VIPs) have been employed to filter the variables that contributed the most to samples clustering. All statistical analyses have been performed using R Studio (Version 1.4.1106, © 2009–2021 RStudio PBC).

## Figures and Tables

**Figure 1 plants-11-01973-f001:**
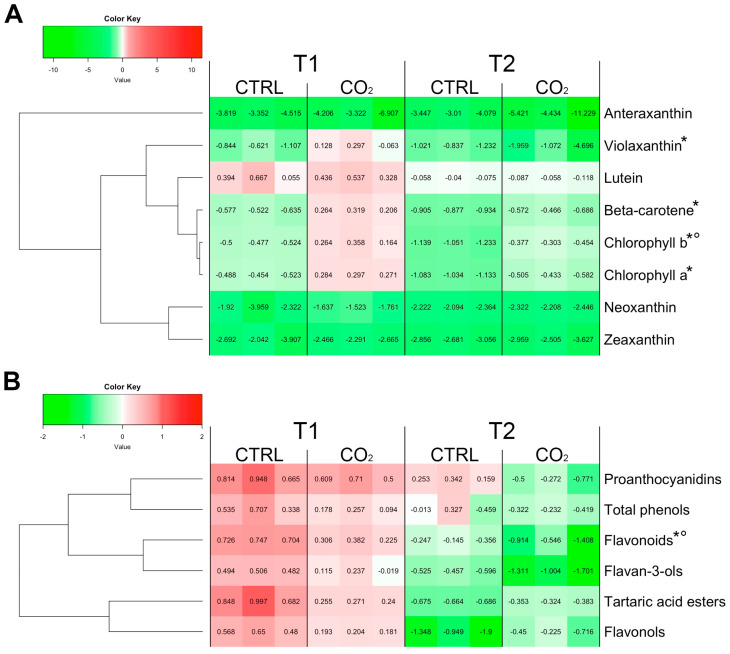
Carotenoid, chlorophyll, xanthophyll (**A**) and phenolic compounds (**B**) content expressed as fold change of their level in correspondence of a weight loss of about 30% (T1) and 45% (T2) in CO_2_ treated and control berry skins. Values at T1 and T2 have been normalized on sample levels at harvest (T0) using the following formula: FC = log_2_ [replicate T1 or T2/mean (T0)]. Every cell of the figure represents the FC value for a specific replicate (analyses have been run in triplicates). A color scale from green to red have been employed to represent FC values in the range from -2 to 2. The symbols ‘*’ and ‘°’ indicate statistically significant differences (*t*-test, *p ≤* 0.05) between control and treated samples at T1 and T2, respectively.

**Figure 2 plants-11-01973-f002:**
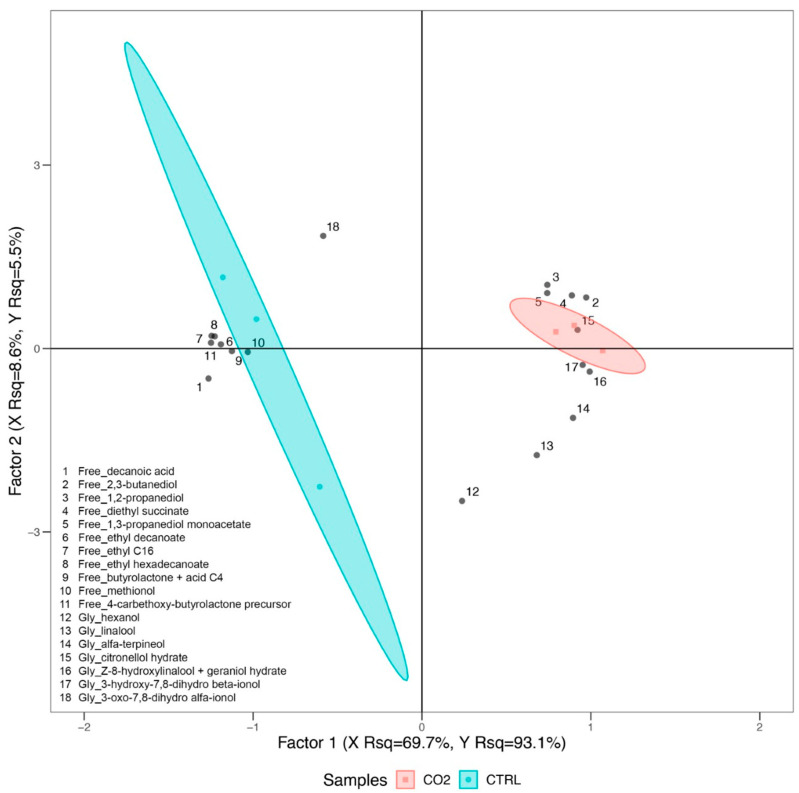
Partial Least Squares Discriminant Analysis (PLSDA) of free and glycosylated compounds in wines. Samples obtained from both control and CO_2_-treated grapes have been analyzed, with molecules that contributed the most to control and treated samples discrimination and that were commonly identified in all of these that have been employed as predictor variables, while treatment has been used as response variable. Variable in projection scores (VIPs) have been used to filter the variables that contributed the most to samples clustering.

**Table 1 plants-11-01973-t001:** Weight loss of Trebbiano berries. Weight loss (% of fresh weight) of the control and CO_2_-treated berries during 73 days of controlled postharvest dehydration. Different letters indicate statistically significant difference (*t*-test, *p ≤* 0.05).

Days	0	15		22		32		36		70		73	
		Control	Treated	Control	Treated	Control	Treated	Control	Treated	Control	Treated	Control	Treated
%	0	21.14 de	16.40 e	22.11 de	25.52 de	23.11 de	28.26 cde	25.87 de	30.76 cde	34.73 bc	44.38 abc	45.78 ab	47.08 a

## Data Availability

Not applicable.

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
