# Peer review of "Short-Term CO2 Treatment of Harvested Grapes (Vitis vinifera L., cv. Trebbiano) before Partial Dehydration Affects Berry Secondary Metabolism and the Aromatic Profile of the Resulting Wine"

_plants, 2022, doi:10.3390/plants11151973_

Round 1

Reviewer 1 Report

The manuscript is clearly written to present the result and data. The analytical data for compounds cover a broad range of categories to give a complete picture of the outcome of CO2 treatment. Here are a few points for the authors to consider in editing the manuscript:

1. Could the mechanism of CO2 impact be discussed or speculated in relationship to the changes of compounds resulted in?

2. Since there is no sensory data, it would be valuable to add some information on the thresholds of some typical compounds, e.g. esters and volatile phenols to show how the scale of changes might impact the sensory properties of the wine?

3. Suggest spell out the acronyms when used first time. e.g. VOC ( Line 129) and RH ( L193).

Author Response

Reviewer 1

Comments and Suggestions for Authors

Comment/Question (C/Q): The manuscript is clearly written to present the result and data. The analytical data for compounds cover a broad range of categories to give a complete picture of the outcome of CO2 treatment.

Answer (A): Thanks for the good evaluation.

C/Q: Here are a few points for the authors to consider in editing the manuscript:

  1. Could the mechanism of CO2 impact be discussed or speculated in relationship to the changes of compounds resulted in?

A: We discuss the CO2 treatment impact on the different classes of compounds in relation to the collected data and the published literature. However, based on our results and observations we do think that going further in detail discussing CO2 mechanisms implies the risk of being too speculative.

  1. Since there is no sensory data, it would be valuable to add some information on the thresholds of some typical compounds, e.g. esters and volatile phenols to show how the scale of changes might impact the sensory properties of the wine?

A: We included odor threshold (OTH) information in the supplementary tables, and we recognize its importance in making molecules impacting the aroma and flavor of grape and wine. However, our research approach has been addressed more to elucidate/report the effect of CO2 on specific chemical classes than the overall effect on the organoleptic characteristics.

  1. Suggest spell out the acronyms when used first time. e.g. VOC (Line 129) and RH ( L193).

A: Thanks for the suggestions, we have modified the text accordingly.

Reviewer 2 Report

Dear Authors,

I revised the paper “Short-term CO2 treatment of harvested grapes (Vitis vinifera L., cv. Trebbiano) before partial dehydration affects berry secondary metabolism and the aromatic profile of the resulting wine”  with pleasure.

The introduction is well written and methods are sound.

I suggest improving the results section. Please, address the following issues:

-         -  I suggest moving tables S1, S2 and S3 from “supplementary material” to the main body, since they are a crucial part of the study.

-          - Table S1: it is not clear id phenolic content is expressed as “µg/g f.w.” or “mg/d dry weight” or “mg GAE/g (line 319). Please, amend.

-          - Table S1, S2 and S3: please, add in the caption the meaning of superscript letters.

-         -  Table S3: please, mind that superscript letters are missing. Please, explain why.

I also suggest adding a conclusion section where strength and limit of the study are reported.

Author Response

Reviewer 2

Comments and Suggestions for Authors

Comment/Question (C/Q): Dear Authors, I revised the paper “Short-term CO2 treatment of harvested grapes (Vitis vinifera L., cv. Trebbiano) before partial dehydration affects berry secondary metabolism and the aromatic profile of the resulting wine” with pleasure. The introduction is well written and methods are sound.

Answer (A): Thanks for the good evaluation.

C/Q: I suggest improving the results section. Please, address the following issues:

  1. I suggest moving tables S1, S2 and S3 from “supplementary material” to the main body, since they are a crucial part of the study.

A: We thank Reviewer 2 for the suggestion. We understand the importance of the absolute values and, in fact, we included this information in the supplementary materials. However, since our approach throughout the manuscript is mainly comparative (CO2 treated vs control), we think that reporting the fold change values (heatmaps analysis in figures 1, as well as the PLSDA analysis reported in figure 2) is clearer and easier for an immediate understanding of the treatment effects. Therefore, we would like to keep the tables in the Supplementary material.

  1. Table S1: it is not clear id phenolic content is expressed as “µg/g f.w.” or “mg/d dry weight” or “mg”GAE/g (line 319). Please, amend.

A: Thanks for the comment, we have reported this information in the caption of the table and in material and method section.

  1. Table S1, S2 and S3: please, add in the caption the meaning of superscript letters

A: Thanks for the comment, we have included this information in the captions.

  1. Table S3: please, mind that superscript letters are missing. Please, explain why.

A: Thanks for the comment, we have included this information in the caption.

  1. I also suggest adding a conclusion section where strength and limit of the study are reported.

A: Thanks for the comment, we have added a conclusion section as suggested.

Reviewer 3 Report

Here are my observations for the paper entitled Short-term CO2 treatment of harvested grapes (Vitis vinifera L.,  cv. Trebbiano) before partial dehydration affects berry secondary metabolism and the aromatic profile of the resulting wine:

Lines 61-63 - citing references are needed

English language must be carefully revised.

You analyzed both the berries and the wines. From my point of view, you should decide weather to keep the results for wines or results for berries. The mixture is not the best option as you do not present a complete description of the final product, which is of real interest for the reader. 

I suggest splitting the information in two research papers: one focusing only the CO2 treatment to berries (individual phenolic analysis, organic acids and GC-MS analyses) and another one focusing the complete characterization of wines by including also the phenolic compounds analysis (and possibly a referring to CO2 treated berries study). 

If you still prefer the manuscript with this structure, at least add comprehensive discussion regarding the influence of berries polyphenols to wine aroma profile. Otherwise, the polyphenols analysis is not justified enough.

Why did you not use selected yeasts? You would have had a more precise control of the experiment. 

Author Response

Reviewer 3 

Comments and Suggestions for Authors

Comment/Question (C/Q): Here are my observations for the paper entitled Short-term CO2 treatment of harvested grapes (Vitis vinifera L., cv. Trebbiano) before partial dehydration affects berry secondary metabolism and the aromatic profile of the resulting wine:

  1. Lines 61-63 - citing references are needed

Answer (A): Thanks for the comment, we have added citing references as suggested.

  1. English language must be carefully revised.

A: Thanks for the comment, the manuscript has been revised and improved where necessary in terms of English language

  1. You analyzed both the berries and the wines. From my point of view, you should decide weather to keep the results for wines or results for berries. The mixture is not the best option as you do not present a complete description of the final product, which is of real interest for the reader. I suggest splitting the information in two research papers: one focusing only the CO2 treatment to berries (individual phenolic analysis, organic acids and GC-MS analyses) and another one focusing the complete characterization of wines by including also the phenolic compounds analysis (and possibly a referring to CO2 treated berries study). If you still prefer the manuscript with this structure, at least add comprehensive discussion regarding the influence of berries polyphenols to wine aroma profile. Otherwise, the polyphenols analysis is not justified enough.

A: Thanks for the comment. We think that when doing research on wine grapes, and so having wine as final product, it is straightforward to work on the effect of a treatment on both the treated samples and on the wine. As stated in the manuscript, our interest addresses both physiological/metabolic modifications induced by CO2 treatment on the berries, and the impact on wine, two aspects that, in our opinion, are and must be strictly connected. In our opinion the analysis of grape polyphenols is well justified by our interest in physiological effect of the treatment, as well as by the fact that previous works report a significant change in berry phenol profile after CO2 treatment. Moreover, at the best of our knowledge, linking the effect of a treatment (CO2 in our case) on grape berry phenols with the wine aroma is a complicated task. If the Reviewer’s suggestion regards the aroma-related phenols, we indeed present data on this chemical class, which appeared to be one of the volatile classes which is affected the most by the treatment, and these results are discussed within the paper.

  1. Why did you not use selected yeasts? You would have had a more precise control of the experiment. 

A: Thanks for the comment. We agree with Reviewer 3: using selected yeasts would have been great since we would have been reducing variability in wine production having a more precise control of the experiment. However, we had to use the standard vinification protocol applied by the winery which was hosting the trial that used natural yeasts of the berries to carry out the fermentation.